# A Novel Functional Electrical Stimulation-Induced Cycling Controller Using Reinforcement Learning to Optimize Online Muscle Activation Pattern

**DOI:** 10.3390/s22239126

**Published:** 2022-11-24

**Authors:** Tiago Coelho-Magalhães, Christine Azevedo Coste, Henrique Resende-Martins

**Affiliations:** 1Graduate Program in Electrical Engineering, Universidade Federal de Minas Gerais, Av, Antônio Carlos 6627, Belo Horizonte 31270-901, MG, Brazil; 2National Institute for Research in Computer Science and Automation (Inria), Camin Team, 34090 Montpellier, France

**Keywords:** Functional Electrical Stimulation, FES-cycling, Reinforcement Learning

## Abstract

This study introduces a novel controller based on a Reinforcement Learning (RL) algorithm for real-time adaptation of the stimulation pattern during FES-cycling. Core to our approach is the introduction of an RL agent that interacts with the cycling environment and learns through trial and error how to modulate the electrical charge applied to the stimulated muscle groups according to a predefined policy and while tracking a reference cadence. Instead of a static stimulation pattern to be modified by a control law, we hypothesized that a non-stationary baseline set of parameters would better adjust the amount of injected electrical charge to the time-varying characteristics of the musculature. Overground FES-assisted cycling sessions were performed by a subject with spinal cord injury (SCI AIS-A, T8). For tracking a predefined pedaling cadence, two closed-loop control laws were simultaneously used to modulate the pulse intensity of the stimulation channels responsible for evoking the muscle contractions. First, a Proportional-Integral (PI) controller was used to control the current amplitude of the stimulation channels over an initial parameter setting with predefined pulse amplitude, width and fixed frequency parameters. In parallel, an RL algorithm with a *decayed-epsilon-greedy* strategy was implemented to randomly explore nine different variations of pulse amplitude and width parameters over the same stimulation setting, aiming to adjust the injected electrical charge according to a predefined policy. The performance of this global control strategy was evaluated in two different RL settings and explored in two different cycling scenarios. The participant was able to pedal overground for distances over 3.5 km, and the results evidenced the RL agent learned to modify the stimulation pattern according to the predefined policy and was simultaneously able to track a predefined pedaling cadence. Despite the simplicity of our approach and the existence of more sophisticated RL algorithms, our method can be used to reduce the time needed to define stimulation patterns. Our results suggest interesting research possibilities to be explored in the future to improve cycling performance since more efficient stimulation cost dynamics can be explored and implemented for the agent to learn.

## 1. Introduction

Functional Electrical Stimulation (FES) is a well-known technique that delivers low-energy electrical pulses through electrodes to evoke muscle contractions in paralyzed or paretic limbs to assist individuals with neurological disorders to reproduce functional movements such as walking, grasping, pedaling, transferring, standing etc. [1]. A popular FES-assisted modality used for rehabilitation is cycling, where the coordinated electrically-evoked contraction of specific muscles within limited ranges of the pedaling cycle allows people with impaired voluntary leg movement to pedal a recumbent tricycle or ergometer [2]. Especially for individuals with Spinal Cord Injury (SCI), when compared to other activities such as standing and walking, FES-cycling has the advantage of minimizing the risk of falls and bone fractures, as the exercise is performed in a recumbent position. Furthermore, it can be accomplished early in the rehabilitation process by individuals with high levels of impairment. The modality has evidenced several physiological and psychological benefits, including positive impacts on pain, cardio-respiratory function, body composition and bone metabolism [3].

Although FES-cycling has been studied for decades and its positive outcomes amply demonstrated, the application of this exercise modality is limited due to reduced efficiency in terms of output power and endurance, as the non-selective responsiveness of electrically evoked contractions favors the early onset of muscle fatigue [4,5]. In this sense, in addition to correctly determine the intervals within the pedaling cycle in which each muscle is recruited, it is therefore necessary to dynamically modulate the pulse parameters (amplitude, width and frequency) to create a synergistic contraction between the different muscle groups, in such a way as to maximize the torque efficiency exerted on the pedals and delay muscle fatigue. In this scenario, the use of open-loop architectures is very limited, and the need for closed-loop control strategies emerges [6].

Several investigations have addressed closed-loop control strategies to track cadence, torque and power [7,8,9]. Besides the performance of the controller and cycling outcomes, the results obtained from investigations indicated that the choice of the strategy to be developed reflects a trade-off between what is expected from the rehabilitation goals and the level of complexity from a technological point of view.

The first assesses whether the use of the technology is aimed at therapeutic applications (cardiovascular conditioning and prevention of atrophy through exercises) that may opt for greater simplicity of implementation in the clinic or related to more complex functional goals, including assistance in activities such as walking, standing, cycling, etc. The second considers the development of technology itself, which addresses the implementation of elaborate electronic instrumentation and the development of complex control strategies and algorithms to deal with non-linearity and temporal variability of muscle response.

On the one hand, simplifying the methods associated with the FES-assisted cycling modality can lead to a drastic reduction in the implementation time and reduce the complexity of the stimulation pattern adjustment, which seems to be a common objective in recent research; on the other hand, the complexity while developing the technology can increase in the background. In any case, however, research possibilities are created for the study of novel methodologies.

For instance, learning control techniques such as iterative learning control (ILC) and repetitive learning control (RLC) have been suggested to automatically adapt the stimulation pattern and improve the performance of tracking repetitive or periodic activities through the use of knowledge obtained in previous interactions between the controller and the system environment [10,11].

In the same context, Reinforcement Learning (RL) emerges as a reward-based machine learning training method whose concept depends on a learner, the decision maker and the so-called agent, who is able to observe the current state of the environment and learn an optimal policy, that is, a stochastic rule by which the agent selects actions as a function of states. In other words, the agent assesses the state of the environment by mapping it into actions and then evaluates a numerical reward obtained from it as training information. A distinguishing feature of RL is, therefore, to formalize the idea of a goal through a reward signal sent from the environment to the agent as a response to the selected action at a specific time. The objective, however, is to maximize the notion of cumulative reward over a specific period and not the immediate reward, which requires the creation of rules for the agent to balance between exploring novel actions and exploiting known actions in the environment [12].

A recent study introduced a method using RL for real-time optimization of stimulation patterns while monitoring the average cadence error during simulated FES-cycling sessions [13]. The performance outcome obtained with a musculoskeletal model simulation was compared to other controllers, such as PID and Fuzzy Logic. Despite not being clear about how the controller would act on the pulse parameters in real scenarios, it was suggested the possibility of a control algorithm to learn how to adapt the electrical pulses of different leg muscles by evaluating the interactions with the system. It was also proposed that the method was able to compensate for muscle fatigue and to track desired cadences. A considered drawback, however, was the large number of interactions and data needed for training the agent. For example, the preliminary results reported the need for an additional ten minutes for training the algorithm before obtaining satisfactory tracking performance, which limited the investigation to being conducted in real-world cycling sessions.

In light of the aforementioned investigations, to reduce the FES-cycling implementation time by simplifying the design of electrical stimulation patterns and reducing the complexity of control techniques, it seems advantageous to use learning methods that reward desired behaviors of a particular agent that, when interacting with the cycling environment, performs trial-and-error actions to learn how to improve the stimulation pattern and consequently the cycling performance.

Whereas generally, in the cycling modality, a baseline stimulation pattern is initially defined so that the pulse parameters can be modulated by a controller, we introduce in this article a new algorithm in which an RL agent is designed to continuously learn through its interaction with the cycling environment how to optimize the same baseline parameters. Through an online quantitative analysis of the cycling cadence error and its relationship with the electrical charge injected to maintain the pedaling movement, the agent modifies the baseline stimulation pattern while a Proportional-Integral (PI) controller modulates the current amplitude to track a pedaling cadence. For the agent to act in the environment and in order to balance between exploring novel actions and exploiting the most rewarding actions, the proposed RL algorithm is based on a *decayed-epsilon-greedy* strategy associated with finite Markov Decision Processes (MDPs), which is used for modeling sequential decision problems.

In the context of FES-cycling, instead of a static stimulation pattern to be modified by a control law, we hypothesized that a non-stationary baseline stimulation pattern would be more efficient in adjusting the amount of injected electrical charge to the time-varying characteristics of the musculature, greatly influencing the stimulation cost during the assisted session. Therefore, from a quick definition of the initial configuration of the stimulation pattern, we intend to assess if the agent would be able to learn to act in the environment aiming to increase the accumulated reward received over time based on a user-defined policy and simultaneously being capable of tracking a preset pedaling cadence.

In addition to suggesting that the use of learning methods would facilitate the implementation of FES-assisted modalities as the complexity of the process of defining the initial stimulation parameters is reduced, we believe that our method has the potential to create interesting research possibilities, such as for example, optimizing the injection of electrical charge to make more efficient the stimulation cost in order to delay the muscle fatigue process.

## 2. Materials and Methods

### 2.1. Subject

One participant (J.S.M., 66 kg, 40 years, male) with complete spinal cord injury (SCI T8, ASI A) for 14 years was recruited for this study and signed a written informed consent. The research was approved by a local Ethical Committee (CAAE: 30989620.6.0000.5149, Ethical Approval number 4.190.128) in agreement with the Declaration of Helsinki. The participant presented unrestricted joint movement and had no cardiovascular problems, epilepsy, or lower limb fractures within the last 12 months prior to the start of the protocol. Before this study began, the volunteer participated in FES-cycling sessions for 18 months.

### 2.2. Materials

Non-stationary FES-cycling sessions were performed overground on a recumbent tadpole tricycle (Arttrike, Porto Alegre, RS, Brazil). The original crankset was replaced by a power-meter crank (2INPOWER, Rotor Bike Components, Madrid, Spain). Orthotic calf support (HASE^®^ Bikes, Waltrop, Germany) was fixed to the pedals to keep the ankle joint at 90° and restrict the leg movement to the sagittal plane. The crank angle was evaluated through an incremental encoder (model LPD3806-360BM-G5-24C). The previously developed constant current topology electrical stimulation device, which is open source and was described in a previously published paper [9], was employed during the sessions. Firmware and software applications (Android app) for the algorithm presented in this paper are available as Appendix A. Data processing and statistical analysis were performed in MATLAB R2021a (MathWorks, Natick, MA, USA).

### 2.3. Methods

#### 2.3.1. Electrical Stimulation

Vastus medialis (VM), vastus lateralis and rectus femoris (VL + RF), and hamstrings (HAM) muscles were stimulated during pedaling sessions. Rectangular self-adhesive surface electrodes (9 × 5 cm, Arktus^®^, Brazil) were used as following settings: (1) an electrode positioned 3 cm above the upper edge of the patella on the distal motor point of the VM muscle; and another electrode on the proximal motor point of the same muscle; (2) an electrode positioned 3 cm above the upper edge of the patella on the distal motor point of the VL muscle and another electrode on the proximal motor point of the RF muscle; (3) one electrode placed 5 cm above the popliteal fossa and the other 15–20 cm above the popliteal fossa for the HAM stimulation. Rectangular and symmetrical biphasic pulses with amplitude limited to 100 mA, a pulse width of 450 μs–600 μs, and a frequency at 35 Hz were employed during the sessions. Pulse amplitude was modulated by a PI controller to track a cadence of 35 rpm chosen to allow the volunteer to cycle for a minimum of 30 min and produce an average power greater than 10 W.

#### 2.3.2. Data Acquisition

A mobile application (Android app) was developed to configure the stimulation parameters and to exchange data wirelessly (Bluetooth) at 10 Hz. The app creates a database for offline processing, which includes data configuration of each channel (ON/OFF, start angle, end angle, frequency, pulse width and amplitude), time and date, reference cadence, actual cadence, and crank angle. It also selects between four different types of cycling modes: (1) manual stimulation; (2) automatic mode without control; (3) automatic mode with control; and (4) RL algorithm enabled. The crank-angle, cadence and power data were also measured by the power meter, which transmitted data via Bluetooth to a personal computer at a sample rate of 50 Hz (manufacturer-specific “fast-mode”). A commercial speedometer was also installed on the tricycle to measure average speed, distance and pedaling time.

#### 2.3.3. Reinforcement Learning Algorithm

In a certain state, the agent acts on the environment for a limited period and expects to receive a numerical reward at the end. In the beginning of every interaction, the agent may choose a different action at a predefined exploratory percentage rate (ε, epsilon). These actions have an impact on the environment, which observation, in turn, leads to a new state. Our algorithm was implemented in such a way that, at a given state, the RL agent may implement a random action to change the baseline stimulation pattern, being to increase or decrease the amplitude and/or pulse width parameters by a predefined value. In this work, however, the pulse frequency and the start and stop angles at which the muscles are stimulated were kept constant.

Within a predefined number of interactions, that is, a sub-sequence of the whole interaction process named episodes, the agent randomly chooses between exploring new actions (increasing/decreasing current amplitude and/or pulse width) or exploiting past actions that have given the highest reward, called greedy actions, to temporarily change the stimulation pattern. The agent then evaluates the rewards received by the temporary implementation of the modifying action on the cycling environment. At the end of every episode, the amplitude and pulse width parameters of each channel are maintained or updated (increased/decreased) according to the greedy action that the agent has learned to be the most rewarding, and a new episode starts from this new baseline setting. Also, at this point, in order to eventually decrease the exploration rate, the epsilon parameter is then multiplied by a decay factor, justifying the *decayed-epsilon-greedy* strategy.

Let $D_{3}=\left\{-0.5,0,1\right\}$ be a predefined numerical set, and the pulse parameters width and amplitude be the variables ω and α, respectively. Let also c={0,1,2,3,4,5}  be the set of six stimulation channels used in the pedaling cycle (VL+RF, VM and HAM; left and right limbs). The set from which the agent can select one possible action is $A=\left\{\left(\alpha ,\omega \right):\alpha ,\;\omega \in D_{3}\right\}$, i.e., the set of all the possible combinations between the two stimulation parameters the agent could modify, totaling nine different actions available independently for each channel. In order to learn which is the most rewarding action within a specific period, the need for exploring different non-greedy actions is created.

In this sense, to balance exploration and exploitation, the *ε-greedy* with a decaying rate dr∈[0,1] method was implemented. As mentioned earlier, at the end of each episode, the ε parameter, which represents the percentage of time the agent will choose to explore new actions, is multiplied by a decaying factor (dr), suggesting that, over time, the agent would be able to find the optimal pattern and the need for exploring changes in these pulse parameters would decrease. These settings depend on how often it is wanted the agent to change the stimulation pattern with the intention of improving it.

Finally, a set of rules was defined as action selection criteria for the agent to follow, including: (1) not exceeding a predefined range of values for the pulse parameters; and (2) not reducing the pulse width when the current amplitude, the variable on which the PI controller acts, is at its maximum value. Increasing the pulse width was allowed in this scenario, however. The process of tuning parameters in the proposed algorithm depends on the project objectives and can be modified to explore different RL approaches associated with FES-assisted activities as well as different strategies for policy and reward implementation. Figure 1 presents the agent-environment interaction scheme.

The interaction dynamics is based on the following description:

The agent interacts with the environment at a sequence of discrete time steps, t=0, 1, 2, … simultaneously for each of the enabled stimulation channels. For a specific time-step t, the agent perceives a state vector St∈ℝ2 consisting of crank angle θ, and average pedaling cadence error e¯(θ˙); and chooses one action for each channel, At,c∈A, which is multiplied by the predefined constants ka and kw. For instance, suppose at time-step t channel 2 has the baseline parameters: a2=45 mA, w2=500 µs and f2=35 Hz. Let us consider the constants defined as ka=1 (mA) and kw=20 (µs). If the agent randomly selects action #5 ({1,−0.5}, see Figure 1), in the next interaction, the channel 2 parameters would be set to 46 mA (45 mA+1×1 mA) and 490 µs (500 µs−0.5×20 µs). Notice that, even if it was not used here, the frequency could also have been included in the algorithm. In this case, considering three variables to be changed (n = 3), the set of possible actions (3n) would be 27. Also, it would be possible to choose to vary the frequency parameter instead of amplitude or width. These possibilities, among the possible inclusion of other parameters (for instance, the stimulation ranges), make the algorithm flexible to suit the purposes defined in many researching and application contexts.The algorithm features a set of rules to which the agent must obey when choosing actions. In this case, limiting values have been set for each of the parameters (α,ω) and was defined in the firmware. For example, for the pulse width starting from the baseline at 500 µs and throughout the cycling session, if the default limiting for the width parameter was set to +100 µs and −50 µs, any chosen action that exceeds 600 µs or is below 450 µs would be refused, and the agent would have to choose another one. It is important to emphasize that this whole procedure is performed by the agent for each channel independently and at every time step.Each interaction impacts on the cycling environment and, from its observation, it is possible to compute a predefined expected reward. In this work, the expected reward was proposed to be defined as described in Equation (1).

(1)$$R_{t+1,c}\left(\dot{\theta },\theta \right)=\{\begin{array}{c} |P_{c}{}^{2}.\sqrt{A_{c}}-{\bar{P_{c}}}^{2}\sqrt{M_{a}-A_{c}})|.\bar{e}{\left(\dot{\theta }\right)}^{-1},\;\;\theta_{c}^{\mathrm{start}}\leq \theta \leq \theta_{c}^{\mathrm{end}}\\ 0,\;\;\;\;\;\;\;\;\;\;\;\;\;\;\;\;\;\;\;\;\;\;\;\;\;\;\;\;\;\;\;\;\;\;\;\;\;\;\;\;\;\;\;\;\;\;\;\;\;\;\;\;\;\;\;\;\;\;\;\;\;\;\;\mathrm{otherwise} \end{array}$$
where $P_{c}=\left(w_{c}+\omega_{c}\;.k_{w}\right).10^{4}$; $A_{c}=(a_{c}+\alpha_{c}.k_{a}+u_{c}).10^{3}$; $\bar{P_{c}}=P_{c,\max}+P_{c,\min}-P_{c}-P_{K}$ (the maximum and minimum values are defined by firmware and $P_{K}$ was set to 100 in our experiment, which was tuned by observing the cycling performance); $M_{a}$ is the maximum amplitude defined at 100 mA; θ is the actual angle within the pedaling cycle and $\dot{\theta }$ is the pedaling cadence; $\bar{e}\left(\dot{\theta }\right)$ is defined as the result of a simple moving average cadence error with window size set to 10; and $\theta_{c}^{\mathrm{start}}$ and $\theta_{c}^{\mathrm{end}}$ correspond to the angle range for the stimulation of channel c. Notice that the rewards are inversely proportional to the cadence error as the agent was designed to exploit actions that promote the lowest cadence error.

The reward dynamics which relate the pulse charge (i.e., the product of pulse width, amplitude and frequency) are demonstrated in Figure 2. It is important to note that, although a constant value for the cadence error was assumed to plot the graph, the rewards obtained are also dependent on the variable value of the cadence error (see Equation (1)). The reward signal can be defined in different ways and in this work it was determined by practical observation of the algorithm performance on the track. After testing some settings and to prevent the pulse amplitude from saturating quickly, it was decided to give greater rewards for larger pulse widths when the amplitude is above half of its range to prioritize the increase of the electrical stimulation pulse charge to sustain muscle contractions longer and, below half of its range value, the rewards are greater for narrower pulse widths aiming to prioritize the stimulation cost.

4.For the agent to decide which action is best, the value of taking each action is therefore needed to be calculated. With the knowledge of the rewards of each chosen action, it is possible to estimate the action values, i.e., the expected return for using action $A_{t}$ in a certain state $S_{t}$. For instance, let us define the expectation of $R_{t}$ given the selection of action $A_{t}$ for any of the possible actions as:



(2)
$$q\left(a\right)≐E\left[R_{t}|A_{t}=a\right]\;\forall \;a\in A$$



The sample-average method was implemented for estimating the action values as defined in Equation (3):(3)$$Q_{n_{a_{c}}+1}=\frac{1}{n_{a_{c}}}\sum_{i=1}^{n_{a_{c}}}R_{i}={\;Q}_{n_{a_{c}}}+\frac{1}{n_{a_{c}}}\left[R_{n_{a_{c}}}-Q_{n_{a_{c}}}\right]$$
where $n_{a_{c}}$ represents the counter of every action for the specific channel. Considering the RL concepts presented in [12], we can think of these estimations as NewEstimate ←OldEstimate + StepSize [Reward−OldEstimate].

5.By maintaining estimates of the action values, it is possible to determine the actions whose estimated values are the greatest (greedy actions). In RL algorithms, the agent exploits its current knowledge of the values of the actions when selecting these greedy-actions. If instead, the agent randomly selects one of the non-greedy actions, then it is exploring because the estimates of the non-greedy action’s value can be improved.

In the proposed algorithm, the simple *decayed-epsilon-greedy* method was implemented to balance exploration and exploitation. This artifice is necessary since exploitation allows to maximize the expected reward in one step, but exploration can yield the greatest total reward in the long run. At every time-step, a random number is created and compared to a predefined epsilon parameter ε ∈[0,1], which refers to the probability of the agent choosing to explore instead of exploiting greedy actions. Different configurations for the ε and decaying-rate parameters can be defined to evaluate the best configuration for the FES-cycling activity. Our hypothesis is that a more exploratory characteristic of the agent can be beneficial for prolonged periods of the FES-assisted cycling activity, as it would learn to adapt the parameters to the real state of the muscle.

6.The algorithm predicts that the interactions between the agent and the environment are broken into sub-sequences called episodes. This is because the baseline parameters also need to be changed in order to find the best stimulation pattern between all channels used. At the end of each episode, the agent chooses the action that returned the greatest action-value and changes the stimulation parameters baseline for all channels. The next episode starts evaluating the agent-environment interaction from this new scenario, therefore independent from the previous one. The user-defined parameter ε is also reduced at the end of each episode.7.Finally, in order to track the desired pedaling cadence, the pulse amplitude of the different channels is modulated simultaneously by a PI controller over the instantaneous baseline stimulation pattern, which is constantly modified by the RL agent. The controller was described in previous work [9]. Thus, even though the agent is responsible for optimizing the stimulation pattern, the PI controller modulates the pulse amplitude for cadence tracking.

At this stage, we do not intend to compare the controller performances but rather to demonstrate that the agent has the ability to learn a predefined policy and that this new approach has some potential to be applied to assisted modalities in order to: (1) overcome the time-consuming problem of stimulation pattern definition; and (2) achieve more efficiency in terms of the injected electrical charge and consequently delay muscle fatigue. Stimulation cost and reward over time, pedaling distance, average speed and power generated on pedals were evaluated.

#### 2.3.4. Intervention Protocol

Three scenarios were considered for the controller: (i) PI control without the RL algorithm enabled; (ii) PI control with the RL algorithm parameters ε=0.6 and dr=0.99 (60%@.99); (iii) PI control with the RL algorithm parameters ε=0.4 and dr=0.99(40%@.99). Each of these three scenarios was evaluated in two different gear ratios for the tricycle, M1 and M2, with the latter being heavier. In all 6 possible scenarios, all stimulation channels started at parameters set to 45 mA and 50 mA for the right and left leg, respectively, pulse width to 500 µs and frequency to 35 Hz. The range for the pulse parameters was set to Δa=10 mA and Δω=100 µs. The parameters $P_{c,\max}$ was therefore set to 600 µs; however, $P_{c,\min}$ was set to 450 µs. The step size for each variable was defined to a=2 mA and ω=20 µs. The agent was supposed to interact with the environment for a predefined period of 5 s before choosing an action once again. Each episode lasted for 45 s before decreasing ε and the modification of the baseline parameters with each channel’s greedy action.

Before the start of each cycling session, the tricycle tires were calibrated and the same electrode placement was respected to ensure similarity between the experiments. The participant was installed on the tricycle to pedal on an official outdoor athletics track. A 3-min of passive cycling on a stationary bike stand (without stimulation) was performed before and after each session. When starting the cycling session, the experimenter only helped the pedal to come out of inertia. The session was limited to sixty minutes and, in case of exceeding that time, until completing the current lap. If the volunteer could not complete the established time and could no longer pedal, the session would last until the tricycle stopped. No human intervention was allowed during the session other than the provision of water at the request of the volunteer while maintaining pedaling.

## 3. Results

A novel algorithm for optimizing the pulse parameters of six different channels based on reinforcement learning concepts was developed for simplifying the control techniques and adjustment of the stimulation parameters to the time-varying characteristics of the non-linearity and time-variability of the muscle dynamics under electric-evoked contractions. A general controller was designed to track a reference cadence and was constituted by a PI-controller acting over a non-stationary stimulation pattern modified by an RL agent. The agent was designed to interact with the cycling environment according to a predefined policy and was supposed to learn to modulate the electrical charge applied to the stimulated muscle groups in order to adjust the injected electrical charge. We evaluated the performance of the proposed algorithm during real cycling experiments. The RL algorithm is presented in Figure 3. An external link for the application firmware and software is available as Appendix A.

Table 1 presents the cycling results obtained from each session configuration and includes distance covered, time duration, average speed and average power. These results are complemented by Figure 4 and Figure 5. Both the proposed general controller, including the RL algorithm and the PI-only controller, allowed the participant to perform overground cycling.

Figure 4 shows the algorithm performance obtained during the six different sessions, divided into M1 (left plots) and M2 (right plots) gear ratio settings. The subplots are divided into: (1) cadence tracking; (2) the sum of the instantaneous amplitude and pulse width of the six stimulation channels; (3) the sum of the injected electrical charge of the six stimulation channels. 

In M1 scenario, despite the “@40%.99” curve being shorter in the plot, it presented a better average speed (4.5 km/h) and managed to cover a greater distance (4688 m) in less time (1 h 02 min 22 s). The “@60%.99” setting has achieved equivalent cycling results in comparison to the PI-only, being, respectively, 4682 m and 4676 m; 1 h 05 min 27 s and 1 h 04 min 59 s; 4.2 km/h and 4.3 km/h; and 10.47 ± 1.89 W and 10.73 ± 1.81 W. It will be interesting in the future to analyze the influence of cadence and gear ratio on cycling performance. So far, however, while the M2 setup is heavier for the rider, the results obtained with the RL suggest a better adaptation to the cycling scenario compared to the PI-only format.

In both M1 and M2 scenarios, it can be seen that the algorithm was able to keep track of the preset pedaling cadence of 35 rpm at least until amplitude saturation. For the M1 setting, the best result was obtained when the algorithm was configured to a 40% exploration rate, in which the controller was able to track the setpoint cadence for the whole session. As the tracking results when the RL algorithm was enabled are very similar to the PI-only configuration for the @60%.99 setting, it can be stated that the algorithm allowed for cadence tracking as designed and the modulation of the stimulation pattern by the RL agent did not negatively impact pedaling performance. It can also be suggested that the pedaling cadence error is more affected when the exploratory rate is higher (i.e., 60%) as the agent is supposed to explore new actions more frequently.

The plot of the second line shows the variation in pulse amplitude and width during the sessions. For the PI-only setting, the pulse width was maintained constant as the controller modulated only the pulse amplitude. It can be noted that the cadence tracking starts to be affected when the amplitude saturates at 100 mA for any of the different scenarios. However, especially for the M2 configuration, the increase in pulse width performed by the agent suggests that the time at which saturation occurred was delayed, as a consequence of the defined policy for his interaction with the cycling environment.

The plots at the bottom show the sum of the injected electrical charge over time for the M1 and M2 settings, considering all channels enabled at that moment. For better visualization, a simple moving average with a window size of 100 was employed. During M2 sessions, it can be noted that the electrical charge reaches higher values more quickly because this scenario is heavier for cycling. However, due to the increase in pulse width performed by the agent, it is observed that the amplitude took longer to reach 100 mA compared to the PI-only configuration. However, the PI-only controller led to an overall lower charge injection. In this work, the electrical charge employed has a direct relationship with the policy defined for the agent to learn. Therefore, as it was defined for larger pulse widths to be more rewarding when the pulse amplitude is above half of its range, then the stimulation cost will be higher for heavier cycling scenarios.

Figure 5 shows the accumulated reward over time obtained by the agent during cycling sessions based on different exploratory rates and the accumulated absolute value of cadence error over time sampled at 10 Hz. It can be seen that the agent is capable of increasing the accumulated reward received by interacting with the environment. However, when the pulse amplitude saturates and the cadence error increases as the controller cannot track the cadence efficiently, the rewards received start to decrease. At this stage, the agent is not able to identify which action is best to implement anymore. During the session “M1: P1 + RL: 60%;@.99“ the application closed unexpectedly, however, without affecting the interpretation of the data. The session duration seems to be shorter compared to Table 1 due to the eventual loss of packets between the mobile and the stimulation system.

## 4. Discussion

In this work we presented an RL algorithm based on a simple *decayed-epsilon-greedy* method to explore different pulse parameter variations with the objective of facilitating the implementation of an optimal stimulation pattern during FES-cycling sessions. While a PI controller was used to track a reference cadence, the RL agent interacted with the cycling environment and learned through trial-and-error how to modulate the muscle activation pattern in order to establish a more efficient stimulation cost.

Figure 4 presents the results for cadence tracking obtained in the six different explored configurations. For the M1 gear ratio, the controller was able to track the 35 rpm cadence longer if compared to the M2 format, as it is less difficult for the rider to cycle. The policy for the agent to learn is associated with the injected electrical charge and the cadence error. When the sessions started, it was observed that the agent prioritized increasing the pulse width. Specially for the M2 setting, this feature delayed the amplitude saturation, resulting in a longer capacity of the algorithm to track the predefined cadence. Indirectly, this feature can positively affect pedaling performance by increasing the average speed and session time.

For instance, for the M2 gear ratio configuration, in which the difficulty for pedaling was greater and it was eventually expected more electrical charge to continue to evoke contractions during the sessions, the RL results suggest a better performance of covered distance and time. When the RL algorithm was enabled, the distance covered by the cyclist was ~31% longer (3733 m for the @60%.99 and 3844 m for the @40%.99 scenarios) compared to sessions where only PI control was used (2931 m). However, it cannot be stated that the PI-only configuration had a worse performance because it was performed some weeks before testing the RL + PI algorithm as a baseline reference and not for comparison purposes. It is possible that the athlete has presented a better result as a result of the training. Therefore, it is important to emphasize that we are not concerned in comparing the controller performances at this stage but instead demonstrate that this new approach has some potential to be applied to FES-assisted modalities.

Our experiments evidenced that the agent correctly learned how to accumulate the received reward. As the goal of the agent is to maximize this quantity over time and not the immediate reward, which justifies the exploratory feature of the algorithm, the agent was supposed to choose to explore novel actions more often when higher exploratory rates were defined. Figure 5 evidences a higher reward accumulated for the 60%-exploration rate during the sessions in the M1 scenario until he was able to track the predefined cadence. For M2, the most exploratory setting returned more reward to the agent in the long term if compared to the 40%-exploration rate as supposed. It can also be seen that, by incrementing the pulse width, the amplitude took longer to saturate (Figure 4) supporting the hypothesis that the agent was able to learn the policy. For session configuration “M1: P1 + RL: 60%;@.99“, a loss of data transfer between the mobile application and the FES system affected the final curve plotting.

Notice that the rewards received by the agent are still related to the pedaling cadence error. In this case, when the amplitude parameters saturate, resulting in the PI controller not being able to guarantee the cadence tracking, a decay in the accumulation of the reward received over time is noticed. This characteristic can be seen in Figure 5 when the accumulated reward curves start to decrease, mainly for the M2 configuration. Also, considering that the agent takes time to interact with the environment to learn how to modify parameters, if the reward relationship favored a decrease in pulse width, it would be expected that the amplitude would be increased by the controller in order to continue tracking the cadence. If the amplitude parameter saturates (considering the amplitude as being the stimulation control output signal), the cadence error would tend to increase, since the controller would not be able to track the cadence anymore. Thus, as the rewards received by the agent are linked to the cadence error, there would be no guarantee for him to correctly estimate the action-values. Therefore, the agent would not be able to recover his capacity to modulate the stimulation pattern correctly.

When comparing the two gear ratio configurations, it can be noticed from Figure 4 that the electrical charge spent during M1 sessions is lower as the output power produced (Table 1), a characteristic consistent with the difficulty of the movement being less than in M2. In the case of lesser pedaling difficulty (scenario M1), where a lower stimulated cost is expected, the configuration with the PI controller just proved to be as efficient. The M2 pedaling configuration seemed more efficient as the pedaling performance has shown better distance covered and session-duration results.

For the two gear ratio configurations, the electrical charge is higher when using the RL algorithm, however. This feat was expected, and it is justified because the reward dynamics were defined here to favor greater pulse widths for greater amplitudes, and from practical observation of the difficulty for our pilot to cycle on the track, it required greater pulse intensities. In this sense, an essential requirement of the algorithm is the definition of the relationship of rewards to the electrical charge used for pedaling, since the agent will prioritize the rules defined for such an experiment. The rewards relationship can therefore be modified to prioritize the reduction of the accumulated electrical charge depending on the experiment to be performed. This feature opens a pathway for the study of new interaction-reward dynamics in the future.

For the proposed controller, it was also necessary to evaluate different parameter settings for the RL algorithm (number of interactions, episode duration, exploration rate, time evaluating rewards, step size for pulse width and amplitude etc.) and policies for the reward-interaction dynamic between the agent and the environment, which is part of the process of tuning the algorithm. Also, in addition to the algorithm parameters themselves, the more actions the agent has available to evaluate the longer it takes to learn. In this sense, the time it takes the agent to learn the policy is related to the algorithm settings and to the number of actions available for him to learn. As muscle fatigue is yet a limitation to the use of electrical stimulation to assist functional activities and consequently a limitation to the session duration, depending on how much the volunteer can pedal, the agent will not have enough time to learn. Therefore, it is necessary to balance the time to learn and the possible cycling capacity in terms of duration for the subject.

To overcome this drawback, a better approach would be to use mathematical models to simulate a priori different combinations of parameters, as well as the agent’s exploration and exploitation characteristics to learn the policy. In [13], researchers suggested the RL controller be trained to learn a defined stimulation strategy to cycle at the desired cadences in simulation before evaluating the performance in the real world. They developed a deep RL algorithm to control the stimulation of different leg muscles during FES-cycling sessions and demonstrated the ability of the agent to learn from its iterations with the system. They trained the agent and compared the results to other controlling methods (PID and Fuzzy Logic Control) using a musculoskeletal model. The results suggested that the proposed method was able to compensate for muscle fatigue and track desired cadences. Unlike the work cited, in which the RL algorithm was used to track cadence, our approach employed a PI control strategy to track a predefined cadence and the RL agent was responsible for adjusting the baseline stimulation pattern. The accumulation of the two functions by the agent will be studied in future works.

The researchers also conducted a simulation study in which a trained RL controller was transferred to another model with a moderately different seat position. Besides potential results, the researchers found that the RL controller required some additional minutes of interaction before producing good tracking performance. Unlike this approach, however, we demonstrate the possibilities of a control method using reinforcement learning concepts during real overground cycling sessions in being able to track desired cycling cadences. In addition to evaluating the performance of the algorithm in three different control strategies, no additional training was needed. Also, parameter adjustment could be made independently for each session, adjusting for the physiological characteristics of that moment. We also demonstrated the agent’s accumulative reward over time, which permits the understanding of his learning capacity. This feature was not demonstrated elsewhere.

In [14], researchers presented an algorithm to automatically detect stimulation intervals by evaluating the torque measured by a commercial crank power meter installed on an instrumented tricycle and using IMUs. In our work, we used an incremental encoder that limited the continuous calculation of the pedaling cadence since it is a device that generates discrete data. Instead of modifying the parameters based on cadence, an approach in which the power or torque exerted on the pedals can be used to evaluate the agent seems to be more adequate. Furthermore, we hypothesize that the use of IMUs could improve the performance of the agent as more data would be available for cadence assessment. In this sense, the time needed for the agent to learn the policy could be reduced, and the evaluation of different reference cadences could be better explored.

Besides the presented method being also able to produce pedaling, we addressed an automatic adaptation of the muscle activation pattern online during the session, a feature that would favor the modulation of parameters to the time-varying conditions of the muscle state. Another approach for this method would be the inclusion of the torque evaluation exerted on pedals for the definition of the initial stimulation pattern (including the stimulation interval) that is yet to be studied in future works.

## 5. Conclusions

As we stated initially, the search for methods that facilitate the definition of stimulation patterns seems to be a common objective of recent studies. Our method presented exploratory results that suggest the possibility of using reinforcement learning methods to control and adjust the stimulation parameters in order to track the desired cadence. The maintenance of the average pedaling cadence throughout the sessions and the increase in the average distance covered by the volunteer indicate the possibility of delaying muscle fatigue using the proposed algorithm.

Three different control configurations were presented, each one being evaluated in two contexts of pedaling difficulty by a subject with spinal cord injury. In all cases where the RL algorithm was enabled, the subject could cycle as well or better than when only the PI controller was operating. According to the interaction-reward relationship established, it was possible to demonstrate that the agent is capable of learning to increase its rewards over time. In our study, we suggest that such a method may favor an increase in pedaling endurance which, in turn, increases the amount of accumulated electrical charge spent during the sessions. However, if the interaction-reward relationship is altered to suit the purposes of greater electrical efficiency, the agent may also learn to reduce the cost of stimulation.

In this sense, our study suggests the possibility of an RL algorithm being able to generate a positive impact on the efficiency of electrical stimuli during FES-assisted cycling sessions since the electrical charge spent to maintain a predefined cadence can be therefore reduced. However, it is necessary that different reward strategies are evaluated together with the policies created for the agent.

Future work will evaluate how the dynamic change of the baseline stimulation pattern in order to achieve a more efficient stimulation cost impact for muscle fatigue and how this method can be used to increase cycling performance. Also, instead of modulating the pulse width and amplitude, we will use this approach to automatically determine the stimulation intervals in order to reduce the implementation time of the FES cycling modality.

## Figures and Tables

**Figure 1 sensors-22-09126-f001:**
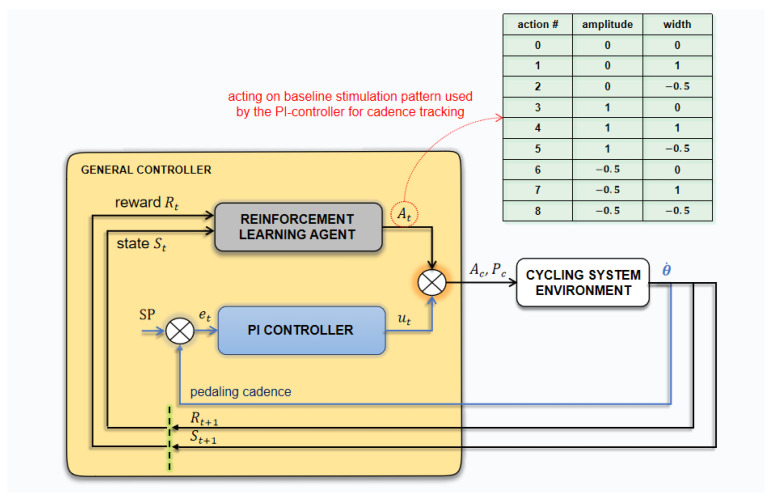
The global control strategy with the agent–environment interaction scheme and the PI-controller. At a state St, the RL agent selects one action for each channel that may change the pulse amplitude and/or width, respecting predefined rules. In parallel, the control contribution from the PI controller for cadence tracking. Signals $A_{c}$ and $P_{c}$ represent the amplitude and pulse width output of the general controller.

**Figure 2 sensors-22-09126-f002:**
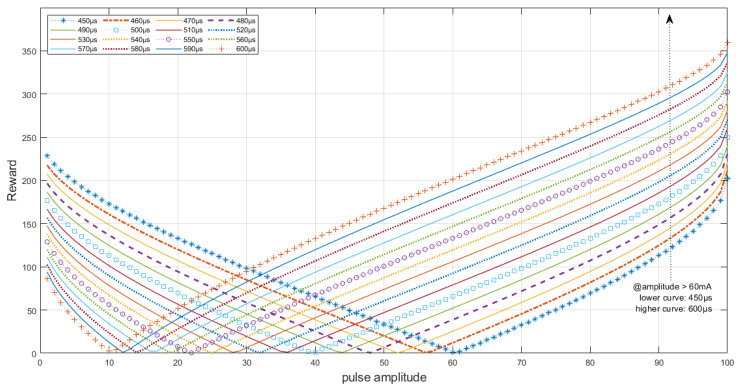
Reward curves relating amplitude and pulse width prioritization. At lower amplitudes, narrower pulse widths return higher rewards as it is preferable to decrease the stimulation cost; at higher amplitudes, the agent is expected to perceive greater rewards as the pulse width increases. The reward must also be related to the average cadence error, however.

**Figure 3 sensors-22-09126-f003:**
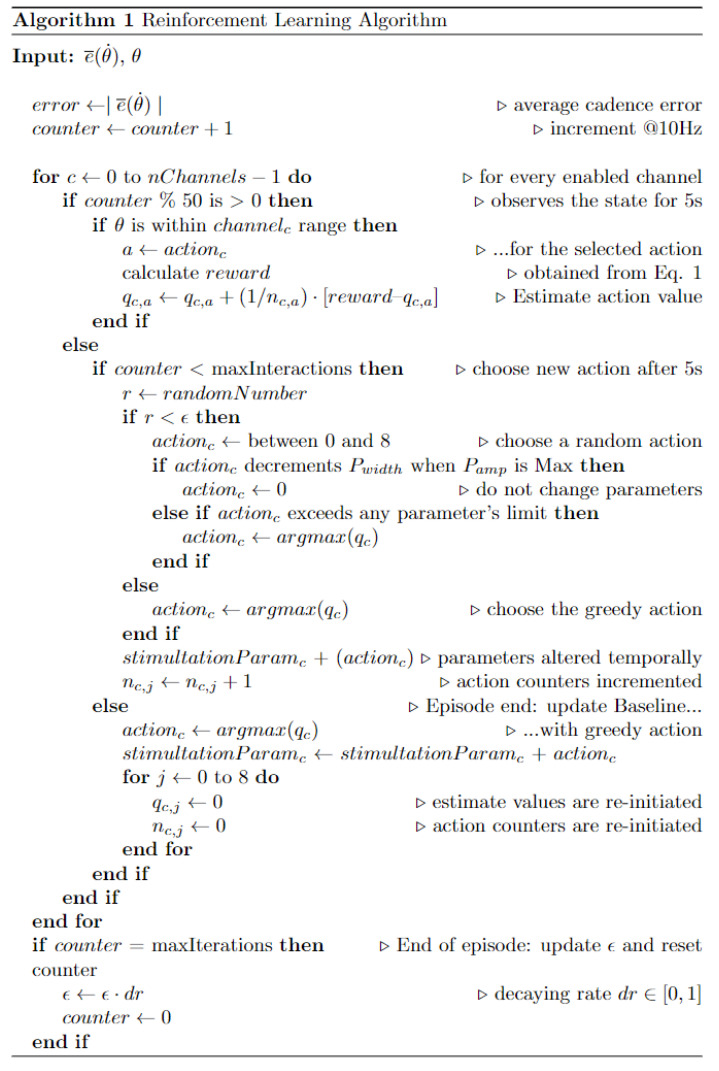
Principle behind the implemented *decayed-epsilon-greedy* RL Algorithm.

**Figure 4 sensors-22-09126-f004:**
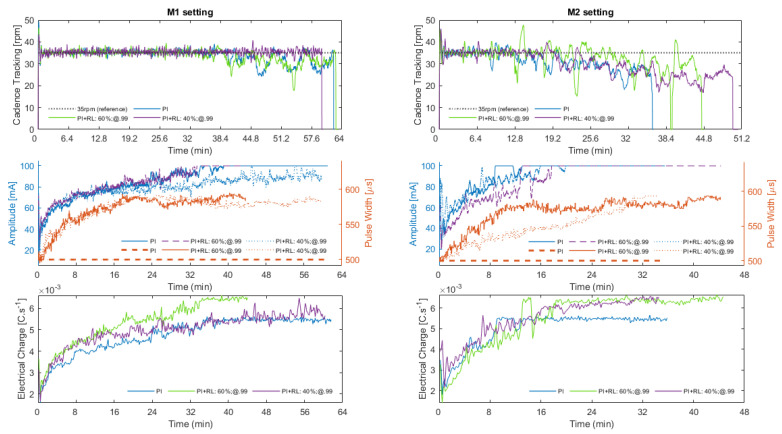
Algorithm performance for M1 (left plots) and M2 (right plots) settings for PI-only, 60% exploration rate and 40% exploration rate configurations. The results presented in subplots are divided into: (1) cadence tracking; (2) modulation of amplitude and pulse width; (3) injected electrical charge. for the six cycling sessions. During the session “M1: P1+RL: 60%;@.99“ the application closed unexpectedly, however, without affecting the interpretation of the data.

**Figure 5 sensors-22-09126-f005:**
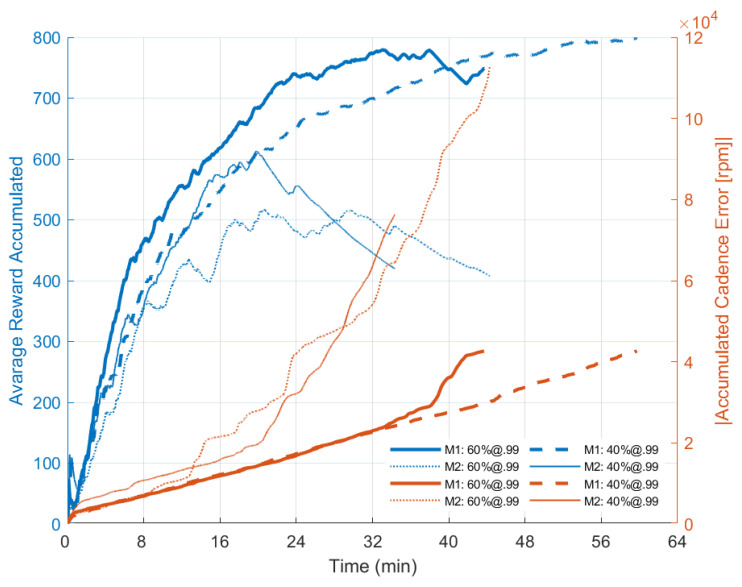
Reward over time during cycling sessions and the accumulated absolute value of cadence error. The agent’s ability to explore different actions over time to increase the reward received can be interpreted as its ability to learn the best action for the time-varying scenario of FES-cycling. Data were obtained from the Android App with a sample frequency @10 Hz.

**Table 1 sensors-22-09126-t001:** Performance results for 6 different control techniques, using or not the RL algorithm. For the PI + RL format, “60%;@.99” means the agent is exploring new actions 60% of the time and decaying this percentage to 99% at the end of every episode. The “40%;@.99” format, in turn, represents a 40% exploratory rate, meaning the agent may choose to exploit actions most of the time instead of exploring new actions.

Session	Format	RL Setting	Gear Ratio	Distance (m)	Duration	Avg. Vel. (km/h)	Avg. Power (W)
1	only PI	NA	M1	4676	1:04:59	4.3	10.73 ± 1.81
2	PI + RL	60%;@.99	M1	4682	1:05:27	4.2	10.47 ± 1.89
3	PI + RL	40%;@.99	M1	4688	1:02:22	4.5	11.53 ± 1.87
4	only PI	NA	M2	2931	0:37:46	4.6	12.18 ± 2.84
5	PI + RL	60%;@.99	M2	3773	0:46:35	4.8	12.90 ± 3.02
6	PI + RL	40%;@.99	M2	3844	0:51:54	4.4	11.33 ± 3.17

## Data Availability

Data sharing is not applicable to this article.

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
