# Peer review of "A Novel Functional Electrical Stimulation-Induced Cycling Controller Using Reinforcement Learning to Optimize Online Muscle Activation Pattern"

_sensors, 2022, doi:10.3390/s22239126_

Round 1

Reviewer 1 Report

This is a well-written paper, which describes a new approach to implement closed-loop FES cycling using reinforcement learning. Although the work is preliminary, it is quite promising and the authors should build upon it by testing it in more participants. The introduction is quite long and can be made concise by removing some of the details regarding fatigue, muscle non-linearity, etc.

Author Response

Response to Reviewer 1 Comments

Comments and Suggestions for Authors: This is a well-written paper, which describes a new approach to implement closed-loop FES cycling using reinforcement learning. Although the work is preliminary, it is quite promising and the authors should build upon it by testing it in more participants. The introduction is quite long and can be made concise by removing some of the details regarding fatigue, muscle non-linearity, etc.

Response: Dear reviewer, I appreciate your valuable comments. The introduction was reduced in ~40% following your suggestion.

Reviewer 2 Report

Good paper showing detailed analysis on use of RL algorithm impacting positively on the efficiency of electrical stimuli during FES-assisted cycling. Results and presentation styles are in good quality. Part of the introduction could be rearranged into a separate literature review section.                          

Author Response

Response to Reviewer 2 Comments

Comments and Suggestions for Authors: Good paper showing detailed analysis on use of RL algorithm impacting positively on the efficiency of electrical stimuli during FES-assisted cycling. Results and presentation styles are in good quality. Part of the introduction could be rearranged into a separate literature review section.                          

Response: Dear reviewer, I appreciate your valuable comments. The introduction was reduced in ~40% following your suggestion.

Reviewer 3 Report

This manuscript presents a novel controller based on a Reinforcement Learning algorithm for real-time adaptation of the stimulation pattern during FES-cycling.

1.      Abstract: Should be compact. Quality results should be provided.

2.      Caption of figures is so long, e.g., figures 1, 2.

3.      In the proposed method, how to balance exploration and exploitation?

4.      Eq. (2) is repeated two times. Correct order of equations.

Author Response

Response to Reviewer 3 Comments

This manuscript presents a novel controller based on a Reinforcement Learning algorithm for real-time adaptation of the stimulation pattern during FES-cycling.

Dear reviewer, I appreciate your valuable comments. Please, find below the responses for each point.

Point 1: Abstract: Should be compact. Quality results should be provided.

Response 1: The abstract was reduced in ~40%. Results include the developed algorithm (which will be uploaded for download and could be used with our open-source Electric Stimulator - referred in the manuscript); the volunteer's cycling performance results; the algorithm performance and capacity to learn a predefined policy; and reward over time.

Point 2: Caption of figures is so long, e.g., figures 1, 2.

Response 2: Thank you for pointing out. The captions were reduced.

Point 3: In the proposed method, how to balance exploration and exploitation?

Response 3: you can find the detailed explanation from line 273 to 298 (first manuscript version). In line 292, for example, you have:

"In this sense, to balance exploration and exploitation, the ε-greedy with a decaying rate  method was implemented. As mentioned earlier, at the end of each episode, the ε parameter, which represents the percentage of time the agent will choose to explore new actions, is multiplied by a decaying factor (dr), suggesting that, over time, the agent would be able to find the optimal pattern and the need for exploring changes in these pulse parameters would decrease. These settings depend on how often it is wanted the agent to change the stimulation pattern with the intention to improve it."

In line 387 other details regarding the process of balancing between exploration and exploitation are presented (#5 of the interaction dynamics description).

Point4: Eq. (2) is repeated two times. Correct order of equations.

Response 4: Thank you for pointing out. The numering was corrected.